# Ralph Kenna’s Scaling Relations in Critical Phenomena

**DOI:** 10.3390/e26030221

**Published:** 2024-02-29

**Authors:** Leïla Moueddene, Arnaldo Donoso, Bertrand Berche

**Affiliations:** 1Laboratoire de Physique et Chimie Théoriques, CNRS—Université de Lorraine, 54000 Nancy, France; leila.moueddene@univ-lorraine.fr; 2L4 Collaboration, Leipzig-Lorraine-Lviv-Coventry, Europe; 3Centre for Fluid and Complex Systems, Coventry University, Coventry CV1 5FB, UK; 4Department of Experimental Physics, Maynooth University, R51 A021 Maynooth, Co. Kildare, Ireland; arnaldoivic2008@gmail.com

**Keywords:** critical exponents, logarithmic corrections, scaling and renormalization, scaling laws, tricritical point, finite-size scaling

## Abstract

In this note, we revisit the scaling relations among “hatted critical exponents”, which were first derived by Ralph Kenna, Des Johnston, and Wolfhard Janke, and we propose an alternative derivation for some of them. For the scaling relation involving the behavior of the correlation function, we will propose an alternative form since we believe that the expression is erroneous in the work of Ralph and his collaborators.

## 1. In Memory of Our Friend Ralph Kenna

This paper is dedicated to our friend Ralph Kenna, who passed away on 26 October 2023. Ralph was a close collaborator and a very good friend, and he was the PhD coadvisor of one of the authors of the present paper (LM). A renowned specialist in the study of phase transitions through the partition function zeros, he developed this formalism in difficult cases where critical behaviors are controlled beyond the dominant singularities by logarithmic corrections.

The present work was initiated on the occasion of the ComPhys23 workshop (http://www.physik.uni-leipzig.de/~janke/CompPhys23/ (accessed on 26 February 2024)) organized by W. Janke in Leipzig, and dedicated to the memory of Ralph. The opening talk was intended to highlight some of Ralph’s most important contributions to statistical physics, notably the derivation of scaling laws between the exponents associated with logarithmic singularities in the vicinity of second-order phase transitions. Ralph was a well-known physicist in the statistical physics community, and these new scaling laws were very successful in the field of critical phenomena. It was by revisiting these scaling laws and their derivation that we noted that one of these relations was incomplete and that the work undertaken by Ralph and his coauthors around twenty years ago merited to be completed.

This paper could have been written by Des Johnston or by Wolfhard Janke, the coauthors of Ralph on this topic (and in many more works).

## 2. Scaling Relations and Universal Combinations of Amplitudes: A Short Primer

We assume that the reader is familiar with the notion of critical exponents that describe the singularities of various thermodynamic functions at the approach of a second-order phase transition. Otherwise, we can suggest referring, e.g., to the textbook of Kardar [1].

Standard scaling relations among the universal critical exponents are the following: (1)$$\begin{array}{ccc} 2\beta +\gamma & = & 2-\alpha , \end{array}$$(2)$$\begin{array}{ccc} \beta (\delta -1) & = & \gamma , \end{array}$$(3)$$\begin{array}{ccc} \nu (2-\eta ) & = & \gamma , \end{array}$$(4)$$\begin{array}{ccc} 2-\alpha & = & \nu d. \end{array}$$They are very useful, not only to obtain the values of all six fundamental critical exponents within a universality class from the knowledge of two of them, but also because they allow for the definition of other universal quantities, written as specific combinations of critical amplitudes. Let us show how this works. For that purpose, we first define the amplitudes as they enter the expressions of the leading singular behaviors of thermodynamic quantities in the vicinity of a second-order phase transition: (5)$$\begin{array}{ccc} & & \mathrm{Specific}\;\mathrm{heat}:\;C_{\pm }\left(t,0\right)≃A_{\pm }{\left|t\right|}^{-\alpha }, \end{array}$$(6)$$\begin{array}{ccc} & & \mathrm{Low}\;\mathrm{temperature}\;\mathrm{magnetization}:\;m_{-}\left(t,0\right)≃B_{-}{\left(-t\right)}^{\beta }, \end{array}$$(7)$$\begin{array}{ccc} & & \mathrm{Susceptibility}:\;\chi_{\pm }\left(t,0\right)≃\Gamma_{\pm }{\left|t\right|}^{-\gamma }, \end{array}$$(8)$$\begin{array}{ccc} & & \mathrm{Critical}\;\mathrm{temperature}\;\mathrm{magnetization}:\;m_{c}\left(0,h\right)≃B_{c}{\left|h\right|}^{1/\delta }, \end{array}$$(9)$$\begin{array}{ccc} & & \mathrm{Correlation}\;\mathrm{length}:\;\xi_{\pm }\left(t,0\right)≃\xi_{0\pm }{\left|t\right|}^{-\nu }. \end{array}$$Here, the two arguments of the functions at the l.h.s. are, respectively, $t=\left(T-T_{c}\right)/T_{c}$ and $h=H/T_{c}$, and the indices ± specify the high (t>0)- and low (t<0)-temperature phases, meaning that the field is zero, and the index *c*, on the contrary implies the field behavior at the critical temperature. The symbol ≃ stands for the leading singularity (i.e., the most singular part since there could be regular contributions to the thermodynamic quantities, power-law corrections to scaling, and multiplicative logarithmic corrections, all these being omitted in Equations (Equation 5)–(Equation 9)).

We can also define the singular part of the free energy density in zero field:(10)$$f_{\pm }^{\mathrm{sing}}\left(t,0\right)≃F_{\pm }{\left|t\right|}^{2-\alpha },$$
and since the specific heat is the second derivative of $f^{\mathrm{sing}}$ with regard to *t*, $F_{\pm }$ is not independent since this requires $A_{\pm }=\left(1-\alpha \right)\left(2-\alpha \right)F_{\pm }$. The Lee–Yang edge is another quantity of interest in critical phenomena, and we define
(11)$$h_{\pm }^{\mathrm{LY}}\left(t\right)≃h_{0\pm }{\left|t\right|}^{\Delta }$$
with the so-called gap exponent Δ=β+γ=βδ.

Universality is the observation that some quantities only depend on very general properties, like space dimensionality. The critical exponents are such universal quantities, but the amplitudes are not, although some combinations among them have the property of universality. To make it clear, let us write Widom’s scaling assumption, i.e., the fact that the singular part of the free energy density is a homogeneous function of the scaling fields
(12)$$f_{\pm }^{\mathrm{sing}}\left(t,h\right)=b^{-d}F_{\pm }\left(\kappa_{t}b^{y_{t}}t,\kappa_{h}b^{y_{h}}h\right),$$
where $F_{\pm }\left(x\left(b\right),y\left(b\right)\right)$ is a universal scaling function of its arguments, $y_{t}$ and $y_{h}$ are the RG dimensions of the relevant fields *t* and *h*, and $\kappa_{t}$ and $\kappa_{h}$ are non-universal metric factors that would differ, say, on the square lattice and the triangular lattice in 2d. The amplitudes defined above depend on these metric factors, and this is why they are not universal, e.g., from $C_{\pm }\left(t,0\right)=\frac{\partial^{2}f_{\pm }^{\mathrm{sing}}\left(t,0\right)}{\partial t^{2}}$, setting $b={\left|t\right|}^{-1/y_{t}}$ in the scaling form (Equation 12), one obtains
(13)$$C_{\pm }\left(t,0\right)=\kappa_{t}^{2+\left(d-2y_{t}\right)/y_{t}}{\left|t\right|}^{\left(d-2y_{t}\right)/y_{t}}{\left(\frac{\partial^{2}F_{\pm }\left(x,0\right)}{\partial x^{2}}\right)}_{x=1}.$$This identifies the exponent $\alpha =\left(2y_{t}-d\right)/y_{t}$ and the amplitude $A_{\pm }=\kappa_{t}^{2-\alpha }{\left(\frac{\partial^{2}F_{\pm }\left(x,0\right)}{\partial x^{2}}\right)}_{x=1}$.

The other exponents are similarly defined in terms of $y_{t}$ and $y_{h}$ by very famous relations that we do not repeat here, and the other amplitudes depend on the metric factors as $B_{-}∼\kappa_{h}\kappa_{t}^{\beta }$, $\Gamma_{\pm }∼\kappa_{h}^{2}\kappa_{t}^{-\gamma }$, $B_{c}∼\kappa_{h}^{1+1/\delta }$.

Simple ratios are immediately defined from the fact that the approach to criticality from above and from below is described by the same exponent for a given quantity (except for the magnetization, obviously), for example, in the case of the specific heat $C_{+}\left(|t|\right)≃A_{+}{\left|t\right|}^{-\alpha }$ and $C_{-}\left(-|t|\right)≃A_{-}{\left|t\right|}^{-\alpha^{\prime }}$, where $\alpha^{\prime }=\alpha$. It follows that the metric factors cancel in the ratio, and
(14)$$R_{C}\left(|t|\right)=\frac{C_{+}\left(|t|\right)}{C_{-}\left(-|t|\right)}=\frac{A_{+}{\left|t\right|}^{-\alpha }}{A_{-}{\left|t\right|}^{-\alpha^{\prime }}}\to R_{C}=\frac{A_{+}}{A_{-}}$$
is thus universal. The limit corresponds to the approach to criticality (|t|→0 here) since the combination $R_{C}\left(|t|\right)$ can be temperature-dependent due to the possible presence of different values for the amplitudes of the corrections to scaling, which has not been taken into account in Equations (Equation 5) to (Equation 9). In the same manner, one defines the universal ratios
(15)$$R_{\chi }=\frac{\Gamma_{+}}{\Gamma_{-}},\;R_{\xi }=\frac{\xi_{0+}}{\xi_{0-}}.$$

The scaling relations are other examples of relations that allow the definition of new combinations. For example, the ratio $m_{-}^{2}/\chi_{\pm }$ eliminates $\kappa_{h}$, and $\kappa_{t}$ is then eliminated, thanks to Equation (Equation 1), if we further divide by $C_{\pm }$. There is still an unwanted ${\left|t\right|}^{2}$ dependence that needs to be simplified, and for that purpose, we consider the quantity
(16)$$R_{\pm }^{\left(1\right)}\left(t\right)=\frac{m_{-}^{2}\left(-|t|\right)}{C_{\pm }\left(t\right)\chi_{\pm }\left(t\right){\left|t\right|}^{2}}=\frac{B_{-}^{2}}{C_{\pm }\Gamma_{\pm }}{\left|t\right|}^{2\beta +\gamma +\alpha -2}\to \frac{B_{-}^{2}}{C_{\pm }\Gamma_{\pm }}.$$Thanks to Equation (Equation 1), the fact that all metric factors cancel out in this latter quantity makes the combination $B_{-}^{2}/C_{\pm }\Gamma_{\pm }$ universal. Proceeding the same way, Equations (Equation 1) and (Equation 2) suggest to contemplate the expressions
(17)$$\begin{array}{ccc} R_{\pm }^{\left(2\right)}\left(t\right) & = & {\left.\chi_{\pm }\left(t\right)m_{-}^{\delta -1}\left(-|t|\right)m_{c}^{-\delta }\left(h\right)h\right|}_{h=h_{\pm }^{\mathrm{LY}}\left(|t|\right)}\\ & & =\Gamma_{\pm }B_{-}^{\delta -1}B_{c}^{-\delta }{\left|t\right|}^{-\gamma +\beta (\delta -1)}\to \Gamma_{\pm }B_{-}^{\delta -1}B_{c}^{-\delta }, \end{array}$$
(18)$$\begin{array}{ccc} R_{\pm }^{\left(3\right)}\left(t\right) & = & \frac{\chi_{\pm }\left(t\right)}{\xi_{\pm }^{2-\eta }\left(t\right)}=\frac{\Gamma_{\pm }}{\xi_{0\pm }^{2-\eta }}{\left|t\right|}^{-\gamma +\nu (2-\eta )}\to \frac{\Gamma_{\pm }}{\xi_{0\pm }^{2-\eta }}, \end{array}$$
that reach their respective universal values. Eventually, Equation (Equation 4) leads to consider the following combination
(19)$$R_{\pm }^{\left(4\right)}\left(t\right)=\xi_{\pm }^{d}\left(t\right)f_{\pm }^{\mathrm{sing}}\left(t\right)=\xi_{0\pm }^{d}F_{\pm }{\left|t\right|}^{-d\nu +2-\alpha }\to \xi_{0\pm }^{d}F_{\pm }$$
as universal also.

## 3. From the Universal Combinations of Amplitudes to Scaling Laws among Hatted Exponents

Having the universal combinations of amplitudes at hand, we consider now the case where the critical behavior is described, besides the leading singularities, by multiplicative logarithmic corrections. This may happen, for example, for a system at its upper critical dimension $d_{\mathrm{uc}}$, or in the case of the 2d four-state Potts model, or the 2d disordered Ising model as well. Many examples can be found in Refs. [2,3,4].

Let us first remind the standard definitions of some exponent combinations that will occur below: $\alpha_{c}=\alpha /\beta \delta$, $\gamma_{c}=\gamma /\beta \delta$, $\nu_{c}=\nu /\beta \delta$, $\epsilon_{c}=\left(1-\alpha \right)/\beta \delta$. The logarithmic corrections can appear either in the approach to the critical temperature when the magnetic field is fixed at zero or, on the other hand, right at $T_{c}$ when the magnetic field approaches zero:(20)$$\begin{array}{cc} & h=0,\;t\to 0^{\pm }, \end{array}$$(21)$$\begin{array}{ccc} & & f_{\pm }^{\mathrm{sing}}\left(t,0\right)≃F_{\pm }{\left|t\right|}^{2-\alpha }{\left(-\ln|t|\right)}^{\hat{\alpha }}, \end{array}$$(22)$$\begin{array}{ccc} & & m_{-}\left(t,0\right)≃B_{-}{\left|t\right|}^{\beta }{\left(-\ln|t|\right)}^{\hat{\beta }}, \end{array}$$(23)$$\begin{array}{ccc} & & e_{\pm }\left(t,0\right)≃\frac{A_{\pm }}{\left(1-\alpha \right)}{\left|t\right|}^{1-\alpha }{\left(-\ln|t|\right)}^{\hat{\alpha }}, \end{array}$$(24)$$\begin{array}{ccc} & & \chi_{\pm }\left(t,0\right)≃\Gamma_{\pm }{\left|t\right|}^{-\gamma }{\left(-\ln|t|\right)}^{\hat{\gamma }}, \end{array}$$(25)$$\begin{array}{ccc} & & C_{\pm }\left(t,0\right)≃A_{\pm }{\left|t\right|}^{-\alpha }{\left(-\ln|t|\right)}^{\hat{\alpha }}, \end{array}$$(26)$$\begin{array}{ccc} & & \xi_{\pm }\left(t,0\right)≃\xi_{0\pm }{\left|t\right|}^{-\nu }{\left(-\ln|t|\right)}^{\hat{\nu }}, \end{array}$$
(27)$$\begin{array}{cc} & t=0,\;h\to 0^{\pm }, \end{array}$$
(28)$$\begin{array}{ccc} & & f_{c}^{\mathrm{sing}}\left(0,h\right)≃F_{c}{\left|h\right|}^{1+1/\delta }{\left(-\ln|h|\right)}^{{\hat{\delta }}_{c}}, \end{array}$$
(29)$$\begin{array}{ccc} & & m_{c}\left(0,h\right)≃B_{c}{\left|h\right|}^{1/\delta }{\left(-\ln|h|\right)}^{{\hat{\delta }}_{c}}, \end{array}$$
(30)$$\begin{array}{ccc} & & e_{c}\left(0,h\right)≃E_{c}{\left|h\right|}^{\epsilon_{c}}{\left(-\ln|h|\right)}^{{\hat{\epsilon }}_{c}}, \end{array}$$
(31)$$\begin{array}{ccc} & & \chi_{c}\left(0,h\right)≃\Gamma_{c}{\left|h\right|}^{-\gamma_{c}}{\left(-\ln|h|\right)}^{{\hat{\delta }}_{c}}, \end{array}$$
(32)$$\begin{array}{ccc} & & C_{c}\left(0,h\right)≃\frac{A_{c}}{\alpha_{c}}{\left|h\right|}^{-\alpha_{c}}{\left(-\ln|h|\right)}^{{\hat{\alpha }}_{c}}, \end{array}$$
(33)$$\begin{array}{ccc} & & \xi_{c}\left(0,h\right)≃\xi_{c}{\left|h\right|}^{-\nu_{c}}{\left(-\ln|h|\right)}^{{\hat{\nu }}_{c}}. \end{array}$$We can also define at criticality t=h=0 the logarithmic correction of the correlation function, defining the exponents $\hat{\eta }$ that will play an essential role in the following of this paper:(34)$$G\left(0,0,|r|\right)≃g_{0}{\left|r\right|}^{-(d-2+\eta )}{\left(\ln|r|\right)}^{\hat{\eta }}.$$We mostly use the notations of Refs. [2,5], with quantities (amplitudes and exponents) at the critical temperature defined with the subscript *c*, except for δ in (Equation 29), which is standard according to the terminology fixed by Fisher long ago [6].

Ralph Kenna and his coworkers, Des Johnston and Wolfhard Janke, have established a series of scaling relations [2,3,4] among “hatted exponents”, as Ralph used to call them. Their approach was based on the zeros of the partition function, either the Lee–Yang zeros (in complex magnetic field) or the Fisher zeros (in complex temperature).

Here, we offer an alternative derivation of most of these scaling laws, probably simpler in its approach. Universality assumes that the previous ratios of amplitudes are still universal when multiplicative logarithmic corrections are present, i.e.,
(35)$$R_{\pm }^{\left(1\right)}\left(t\right)=\frac{m_{-}^{2}\left(-|t|\right)}{C_{\pm }\left(|t|\right)\chi_{\pm }{\left(|t|)|t\right|}^{2}}=\frac{B_{-}^{2}}{C_{\pm }\Gamma_{\pm }}{\left(-\ln|t|\right)}^{2\hat{\beta }-\hat{\alpha }-\hat{\gamma }}.$$The fact that this quantity *must* tend to $B_{-}^{2}/C_{\pm }\Gamma_{\pm }$ now demands that
(36)$$2\hat{\beta }=\hat{\alpha }+\hat{\gamma }.$$This is the first of Ralph and his coworkers’ scaling relations. Using the same method, the second ratio easily leads to a second relation:(37)$$\hat{\gamma }+\hat{\beta }\left(\delta -1\right)-\delta \hat{\delta }=0.$$The amplitude of the Lee–Yang edge, $h_{0\pm }$, has a non-trivial dependence with the metric factors, $h_{0\pm }∼\kappa_{t}^{\beta \delta }\kappa_{h}^{-1}$. This can be retrieved from the scaling relation Δ=βδ, and the universality of the ratio
(38)$$R^{\left(5\right)}\left(t\right)=\frac{m_{c}\left(h_{\pm }^{\mathrm{LY}}\left(t\right)\right)}{m_{-}\left(t\right)}=\frac{B_{c}h_{0\pm }^{1/\delta }}{B_{-}}{\left|t\right|}^{\Delta /\delta -\beta }{\left(-\ln|t|\right)}^{\hat{\Delta }/\delta +\hat{\delta }-\hat{\beta }}\to \frac{B_{c}h_{0\pm }^{1/\delta }}{B_{-}}$$
requires that
(39)$$\hat{\Delta }=\left(\hat{\beta }-\hat{\delta }\right)\delta .$$The scaling relations (Equation 36) and (Equation 37) were first derived in Ref. [3]. Instead of (Equation 39), Ralph and his coworkers had $\hat{\Delta }=\hat{\beta }-\hat{\gamma }$, which is recovered here using (Equation 37) and (Equation 39).

In the same paper, they also derived
(40)$$\hat{ϙ}=\hat{\nu }+\nu \hat{\alpha }/\left(2-\alpha \right)$$
that we will now consider. This is an analog of the hyperscaling relation for logarithmic relations since it is derived from the same ratio of amplitudes as the ordinary hyperscaling relations, even though the space dimension, fixed at its upper critical value, does not appear explicitly. A new *pseudo-critical exponent* [3,4,7] appears there, $\hat{ϙ}$, which describes the finite-size scaling (FSS) of the correlation length
(41)$$\xi_{L}≃L{\left(\ln L\right)}^{\hat{ϙ}}.$$This behavior is encoded in the scaling hypothesis for the correlation length, appropriately extended to account for the logarithmic correction: (42)$$\xi_{\pm }\left(t,h,L^{-1}\right)=b{\left(\ln b\right)}^{\hat{ϙ}}X\left(\kappa_{t}b^{y_{t}}{\left(\ln b\right)}^{{\hat{y}}_{t}}t,\kappa_{h}b^{y_{h}}{\left(\ln b\right)}^{{\hat{y}}_{h}}h,bL^{-1}\right).$$Like $y_{t}$ (respectively, $y_{h}$) is the RG eigenvalue associated with the scaling field *t* (respectively, *h*), we denote ${\hat{y}}_{t}$ (respectively, ${\hat{y}}_{h}$) the corresponding exponent of the logarithmic correction. For the sake of clarity, we will later denote the rescaled variables as $x\left(b\right)=\kappa_{t}b^{y_{t}}{\left(\ln b\right)}^{{\hat{y}}_{t}}t$, $y\left(b\right)=\kappa_{h}b^{y_{h}}{\left(\ln b\right)}^{{\hat{y}}_{h}}h$, and $z\left(b\right)=bL^{-1}$. Equation (Equation 41) follows from the choice b=L at criticality t=h=0 in (Equation 42). The same scaling form is used in the thermodynamic limit L→∞, setting x=1. This requires iterations
(43)$$\begin{array}{ccc} b & = & {\left(\kappa |t|\right)}^{-1/y_{t}}{\left(\ln b\right)}^{-{\hat{y}}_{t}/y_{t}}\\ & ≃ & {\left(\kappa |t|\right)}^{-1/y_{t}}{\left(-\ln|t|\right)}^{-{\hat{y}}_{t}/y_{t}}\left(1+\frac{\ln(-\ln|t|)}{\left(-\ln|t|\right)}+\;\mathrm{higher}\;\mathrm{order}\;\mathrm{correction}\right). \end{array}$$Insertion into the expression of the correlation length leads to the leading order
(44)$$\xi_{\pm }\left(t,0,0\right)≃{\left|t\right|}^{-1/y_{t}}{\left(-\ln|t|\right)}^{\hat{ϙ}-{\hat{y}}_{t}/y_{t}}X\left(1,0,0\right)$$
and requires the usual relation $\nu =1/y_{t}$ to conform to (Equation 26): (45)$$\hat{ϙ}=\hat{\nu }+\nu {\hat{y}}_{t}.$$We now show that this agrees with Ralph’s scaling relation (Equation 40). For that purpose, we use the compatibility with the extension of the phenomenological Widom scaling assumption for the free energy density (Equation 12) to the presence of logarithmic corrections, written as far as we know for the first time by Ralph Kenna in Ref. [8],
(46)$$f_{\pm }^{\mathrm{sing}}\left(t,h,L^{-1}\right)=b^{-d}F_{\pm }\left(\kappa_{t}b^{y_{t}}{\left(\ln b\right)}^{{\hat{y}}_{t}}t,\kappa_{h}b^{y_{h}}{\left(\ln b\right)}^{{\hat{y}}_{h}}h,bL^{-1}\right).$$The second derivative with regard to *t* is the specific heat, and the choice x=1 at $h=L^{-1}=0$ then leads (using $\alpha =\left(2y_{t}-d\right)/y_{t}$) to
(47)$$C_{\pm }\left(t,0,0\right)≃{\left|t\right|}^{-\alpha }{\left(-\ln|t|\right)}^{2{\hat{y}}_{t}-\alpha {\hat{y}}_{t}}C_{\pm }\left(1,0,0\right),$$(from now on, we always limit (Equation 43) to leading logarithmic order); hence, from (Equation 25),
(48)$$\hat{\alpha }=\left(2-\alpha \right){\hat{y}}_{t}$$
which completes the proof.

## 4. Solving a Disagreement with Our Friends

In Ref. [2], Ralph Kenna has given a complete account of these and many more scaling relations among hatted exponents. This is not our purpose here to be exhaustive, but rather to show alternative derivations, or to complete what Ralph and his coworkers did not do. With this perspective in mind, Equations (Equation 42) and (Equation 46) and an analogous homogeneity form for the correlation function (discussed later) offer an option to proceed as we show now.

In Ref. [4], two other scaling relations between hatted exponents were derived: (49)$$\begin{array}{ccc} & & \hat{\alpha }=d\left(\hat{ϙ}-\hat{\nu }\right)\;\mathrm{or}\;\hat{\alpha }=1+d\left(\hat{ϙ}-\hat{\nu }\right), \end{array}$$(50)$$\begin{array}{ccc} & & \hat{\eta }=\hat{\gamma }-\hat{\nu }\left(2-\eta \right). \end{array}$$Concerning the first relation (Equation 49), the second formula is valid in such circumstances where the model has α=0 and an impact angle ϕ≠π/4 for the Fisher zeros in the complex plane (this is the case for the pure two-dimensional Ising model). We will not consider this case, but rather the more general case of the first formula. It can be rederived by careful use of the ratio 4 (Equation (Equation 19)) in Section 2, and even requires the use of FSS of the correlation length. From $\xi_{\pm }\left(t\right)≃{\left|t\right|}^{-\nu }{\left(-\ln|t|\right)}^{\hat{\nu }}$, we first reverse to $\left|t\right|≃\xi_{\pm }^{-1/\nu }\left(t\right){\left(\ln\xi_{\pm }\left(t\right)\right)}^{\hat{\nu }/\nu }$. This expression is then incorporated into (Equation 20) to obtain
(51)$$f_{\pm }^{\mathrm{sing}}\left(t\right)≃\xi_{\pm }^{-d}\left(t\right){\left(\ln\xi_{\pm }\left(t\right)\right)}^{d\hat{\nu }+\hat{\alpha }}$$
i.e., a modified version of (Equation 19):(52)$$R_{\pm }^{\left(4bis\right)}\left(t\right)=\frac{\xi_{\pm }^{d}\left(t\right)f_{\pm }^{\mathrm{sing}}\left(t\right)}{{\left(\ln\xi_{\pm }\left(t\right)\right)}^{d\hat{\nu }+\hat{\alpha }}}\to \xi_{0\pm }^{d}F_{\pm }.$$

Now, inserting (Equation 41) into (Equation 51) leads to the FSS behavior of the free energy density at criticality
(53)$$f_{L}^{\mathrm{sing}}\left(0\right)≃L^{-d}{\left(\ln L\right)}^{-d\hat{ϙ}}{\left(\ln L\right)}^{\hat{\alpha }+d\hat{\nu }},$$
and compatibility with (Equation 46) at t=h=0, b=L then demands
(54)$$\hat{\alpha }=d\hat{ϙ}-d\hat{\nu }.$$
which is Kenna and his coworkers’ relation.

The same derivation can be carried out for the magnetic sector, considering the approach to criticality at $T_{c}$ for h→0, yielding the scaling relation
(55)$$\hat{\delta }=d\hat{ϙ}-d{\hat{\nu }}_{c}.$$

Concerning Equation (Equation 50), we believe that this relation is incomplete. Applied to the four-state Potts model in two dimensions [9,10,11,12], which has $\hat{\eta }=-\frac{1}{8}$, $\hat{\gamma }=\frac{3}{4}$, $\hat{\nu }=\frac{1}{2}$, and $\eta =\frac{1}{4}$, Equation (Equation 50) is fulfilled. We believe that this is because $\hat{ϙ}=0$, and that an additional term $\hat{ϙ}\gamma /\nu$ is missing in the general case. A test is provided in the case of the Ising model in four dimensions, which has $\hat{ϙ}=\frac{1}{4}$ (models at their upper critical dimensions have $\hat{ϙ}=1/d_{\mathrm{uc}}$ [7,13,14,15]). There, Ralph and his coauthors had anticipated that $\hat{\eta }=0$ for Equation (Equation 50) to work ($\hat{\gamma }=\frac{1}{3}$, $\hat{\nu }=\frac{1}{6}$, and η=0 for the Ising model in four dimensions), but according to Luijten [16], $\hat{\eta }=\frac{1}{2}$ instead, a result that is in contradiction with Equation (Equation 50).

Let us examine the problem in more detail. In Ref. [4], the authors have questioned the relation between the correlation function and the square of the magnetization when the system decorrelates, i.e., for |r|→∞:(56)$$G(t,h,L^{-1}\to \infty ,|r|\to \infty )\to m^{2}\left(t,h,L^{-1}\to \infty \right).$$On the contrary, we assume that there is no reason why this would not be valid, so we start from the homogeneity of the (spin–spin) correlation function with logarithmic corrections as
(57)$$G(t,h,L^{-1},|r|)=b^{-(d-2+\eta )}{\left(\ln b\right)}^{\hat{\eta }}G\left(x\left(b\right),y\left(b\right),z\left(b\right),|r|/b\right),$$Setting x=1, y=0 and the thermodynamic limit z=0 leads to the following temperature behavior when |r|→∞
(58)$$G\left(t,0,0,\infty \right)≃{\left|t\right|}^{\frac{d-2+\eta }{y_{t}}}{\left(-\ln|t|\right)}^{\frac{d-2+\eta }{y_{t}}{\hat{y}}_{t}+\hat{\eta }}G\left(1,0,0,\infty \right)=m^{2}\left(t,0,0\right).$$This requires the usual relation $2\beta =\frac{d-2+\eta }{y_{t}}$ and
(59)$$\hat{\eta }=2\left(\hat{\beta }-\beta {\hat{y}}_{t}\right).$$The two examples given above are test grounds. For the four-state Potts model in two dimensions, we extract immediately $\hat{\eta }=-\frac{1}{8}$, which is correct. For the 4d Ising model, on the other hand, we obtain $\hat{\eta }=\frac{1}{2}$, in agreement with Luijten’s result [16], later verified numerically in Ref. [17], but we are here in contradiction with the prediction of Refs. [2,4].

Since the question is of importance, we want to consider it from other perspectives also. The correlation function is linked to the susceptibility via the fluctuation–dissipation relation:(60)$$\chi \left(0,0,L^{-1}\right)=\int_{0}^{L}d^{d}{r\;\;|r|}^{-(d-2+\eta )}{\left(\ln|r|\right)}^{\hat{\eta }}G(0,0,|r|L^{-1},1).$$Setting ρ=|r|/L, we have
(61)$$\chi \left(0,0,L^{-1}\right)=L^{2-\eta }{\left(\ln L\right)}^{\hat{\eta }}\int_{0}^{1}d^{d}\rho \;\;\rho^{-(d-2+\eta )}{\left(1+\frac{\ln\rho }{\ln L}\right)}^{\hat{\eta }}G\left(0,0,\rho ,1\right)≃L^{2-\eta }{\left(\ln L\right)}^{\hat{\eta }},$$
and since the susceptibility obeys, via the second derivative of (Equation 46),
(62)$$\chi \left(t,h,L^{-1}\right)=\kappa_{h}^{2}b^{-d+2y_{h}}{\left(\ln b\right)}^{2{\hat{y}}_{h}}Y\left(x\left(b\right),y\left(b\right),z\left(b\right)\right),$$
its FSS compared to (Equation 61) demands that $2-\eta =2y_{d}-d=\gamma /\nu$ and
(63)$$\hat{\eta }=2{\hat{y}}_{h}.$$Again, this confirms $\hat{\eta }=\frac{1}{2}$ for the 4d Ising model.

The fluctuation–dissipation theorem has also been used in Refs. [2,4] in the form
(64)$$\chi_{\infty }\left(t\right)∼\xi_{\infty }^{2-\eta }\left(t\right){\left(\ln\xi_{\infty }\left(t\right)\right)}^{\hat{\eta }},$$
from where Equation (Equation 50) follows, so there is still some difficulty hidden to solve our disagreement. Let us set $b=\xi_{\infty }\left(t\right)$ in (Equation 62):(65)$$\chi_{\infty }\left(t\right)=\xi_{\infty }^{-d+2y_{h}}\left(t\right){\left(\ln\xi_{\infty }\left(t\right)\right)}^{\hat{\eta }}Y\left(x\left(\xi_{\infty }\left(t\right)\right),0\right)$$
where the variable *x* in the scaling function Y is evaluated at $\xi_{\infty }\left(t\right)$ to give
(66)$$x\left(\xi_{\infty }\left(t\right)\right)=\xi_{\infty }{\left(t\right)}^{y_{t}}{\left(\ln\xi_{\infty }\left(t\right)\right)}^{{\hat{y}}_{t}}\left|t\right|={\left(-\ln|t|\right)}^{\hat{ϙ}/\nu }.$$The scaling function must have the behavior $Y\left(x,0\right)∼x^{-\gamma }$ when |t|→0 to recover the temperature singularity of the susceptibility $\chi_{\infty }\left(t,0\right)∼{\left|t\right|}^{-\gamma }{\left(-\ln|t|\right)}^{\hat{\gamma }}$. It follows that instead of (Equation 64), one has
(67)$$\chi_{\infty }\left(t\right)∼\xi_{\infty }^{2-\eta }\left(t\right){\left(\ln\xi_{\infty }\left(t\right)\right)}^{\hat{\eta }-\gamma \hat{ϙ}/\nu }$$
and instead of Equation (Equation 50), one has a third form for the exponent $\hat{\eta }$: (68)$$\hat{\eta }=\hat{\gamma }-\hat{\nu }\left(2-\eta \right)+\gamma \frac{\hat{ϙ}}{\nu },$$
again compatible with $\hat{\eta }=\frac{1}{2}$ for the 4d Ising model. This suggests to use, instead of the ratio $R_{\pm }^{\left(3\right)}\left(t\right)$, the modified version
(69)$$\begin{array}{ccc} R_{\pm }^{\left(3\mathrm{bis}\right)}\left(t\right) & = & \frac{\chi_{\pm }\left(t\right)}{\xi_{\pm }^{2-\eta }\left(t\right){\left(\ln\xi_{\pm }\left(t\right)\right)}^{\hat{\eta }-\gamma \hat{ϙ}/\nu }}\\ & = & \frac{\Gamma_{\pm }}{\xi_{0\pm }^{2-\eta }}{\left(-\ln|t|\right)}^{\hat{\gamma }-\hat{\eta }-\gamma \hat{ϙ}/\nu -\hat{\nu }\left(2-\eta \right)}\to \frac{\Gamma_{\pm }}{\xi_{0\pm }^{2-\eta }} \end{array}$$
which, again, is universal. The four standard scaling laws and the corresponding four hatted scaling laws are listed in Table 1.

So far so good, but the situation is not yet clear since the case of percolation in six dimensions (its upper critical dimension) is maybe a counterexample. With γ=1, $\nu =\frac{1}{2}$, η=0, and the values of the logarithmic correction exponents $\hat{\gamma }=\frac{2}{7}$, $\hat{\nu }=\frac{5}{42}$, and the pseudo-critical exponent $\hat{ϙ}=\frac{1}{d_{\mathrm{uc}}}=\frac{1}{6}$, using Equation (Equation 68), we predict $\hat{\eta }=\frac{8}{21}$, while Kenna and his coworkers predict $\hat{\eta }=\frac{1}{21}$ from Equation (Equation 50). This latter result conforms to an analytic prediction from Ref. [18], but on the other hand, our value is supported by an FSS prediction by Ruiz-Lorenzo [19], $\chi_{L}L^{-2}∼{\left(\ln L\right)}^{8/21}$. This disagreement demands further analysis.

In Ref. [2], Kenna listed the values of the known hatted critical exponents for a series of models, and when $\hat{\eta }$ was not known, he proposed the expected value from the use of Equation (Equation 50). An interesting model is missing from the list, the tricritical Ising model, which has logarithmic corrections at its upper critical dimension $d_{\mathrm{uc}}=3$ and has non-zero $\hat{ϙ}=1/3$. We will now analyze this universality class in more detail.

## 5. The Tricritical Ising Universality Class in the Blume–Capel Model at the Upper Critical Dimension

The spin-1 Blume–Capel model [20,21] is a lattice model defined by the Hamiltonian
(70)$$H=-J\sum_{\langle i,j\rangle }\sigma_{i}\sigma_{j}+\Delta \sum_{i}\sigma_{i}^{2}-H\sum_{i}\sigma_{i},$$
where the spin variables $\sigma_{i}=-1,0,+1$, J>0 denotes the ferromagnetic exchange interaction among nearest-neighbor sites (〈i,j〉 indicates summation over nearest neighbors), and Δ is the crystal-field strength that controls the density of vacancies (the $\sigma_{i}=0$ states can be viewed as vacancies in an ordinary $\sigma_{i}=\pm 1$ Ising model) [22]. For Δ=−∞, vacancies are suppressed from the partition function, and the Hamiltonian reduces to that of the Ising ferromagnet. At Δ=0 the second-order transition is in the pure Ising model universality class. When Δ increases from 0, a perturbation theory shows that the transition temperature decreases along a line that remains of Ising-like second-order phase transition. On the other hand, in the vicinity of T=0, the transition is first order and persists first order at small values of *T* until it reaches the second-order line. Right at the limit, there is a tricritical point characterized by specific values of $T_{t}$, $\Delta_{t}$ (Figure 1). Tricriticality corresponds to the $\phi^{6}$ Landau expansion [23], and the upper critical dimension is thus $d_{\mathrm{uc}}=3$, the case that we consider now.

Using appropriate linear combinations of the physical parameters, determined by the geometry of the phase diagram and the fact that they have to vanish at the tricritical point, the even scaling fields are $\tau =\left(T-T_{t}\right)/T_{t}$, $g=\left(\Delta -\Delta_{t}\right)/T_{t}+a\tau$, and the odd scaling field is the usual magnetic field $h=H/T_{t}$. Lawrie and Sarbach [24] have shown that the free energy density at the tricritical point in 3d reads, in terms of these scaling fields, as
(71)$$f_{\mathrm{tri}}^{\mathrm{sing}}\left(\tau ,g,h\right)=b^{-3}F_{\pm }\left(b^{1}{\left(\ln b\right)}^{\frac{4}{15}}\tau ,b^{2}{\left(\ln b\right)}^{\frac{1}{3}}g\left(1-a^{-1}b^{-1}{\left(\ln b\right)}^{-\frac{1}{15}}\tau g\right),b^{\frac{5}{2}}{\left(\ln b\right)}^{\frac{1}{6}}h\right),$$
where the non-universal metric factors $\kappa_{i}$ have been omitted. Here, the subscript “tri” indicates that the expression is valid in the vicinity of the tricritical point where τ=g=h=0. The logarithmic corrections are explicitly given in this expression.

The notations in Ref. [24] necessitate to be adapted to be consistent with those that we have used until now. Leaving aside the logarithmic corrections for a while, let us write
(72)$$f_{\mathrm{tri}}^{\mathrm{sing}}\left(\tau ,g,h\right)=b^{-d}F_{\pm }\left(b^{y_{\tau }}\tau ,b^{y_{g}}g,b^{y_{h}}h\right).$$The dominant even scaling field is *g*, the usual singularities and critical exponents are therefore defined by their behaviors with regard to *g* instead of τ, which brings corrections to scaling due to crossover. This means that the *t* and $y_{t}$ of the three previous sections of this paper will now be replaced by *g* and $y_{g}$. At h=0 and τ=0, setting $b={\left|g\right|}^{-1/y_{g}}$ leads to $f_{\mathrm{tri}}^{\mathrm{sing}}\left(0,g,0\right)∼{\left|g\right|}^{d/y_{g}}$. This is compatible with $\nu =1/y_{g}$. The specific heat measures the total energy fluctuations. Its most singular part is defined by
(73)$$\begin{array}{ccc} & & C\left(\tau ,g,h\right)=\frac{\partial^{2}f_{\mathrm{tri}}^{\mathrm{sing}}}{\partial g^{2}}=b^{-d+2y_{g}}C_{\pm }\left(b^{y_{\tau }}\tau ,b^{y_{g}}g,b^{y_{h}}h\right)∼{\left|g\right|}^{-\alpha }\;\mathrm{when}\;\tau ,h=0 \end{array}$$
with $\alpha =\frac{2y_{g}-d}{y_{g}}$. This shows that the exponent $d/y_{g}$ of $f_{\mathrm{tri}}^{\mathrm{sing}}\left(0,g,0\right)$ is thus equal to the usual value 2−α, and the hyperscaling relation holds.

We can also define less singular exponents with regard to the scaling field τ (with tilde notation), e.g., $C\left(\tau ,0,0\right)∼{\left|\tau \right|}^{-\tilde{\alpha }}$. A similar analysis as above shows that $f_{\mathrm{tri}}^{\mathrm{sing}}\left(\tau ,0,0\right)∼{\left|\tau \right|}^{d/y_{\tau }}$ with $d/y_{\tau }=\left(d/y_{g}\right)\phi =\left(2-\alpha \right)\phi$ with $\phi =y_{g}/y_{\tau }$ the crossover exponent. There is a caveat here since $d/y_{\tau }$
*is not equal to*
$2-\tilde{\alpha }$, as one can find in the literature [24]. Indeed, $\tilde{\alpha }=\left(2y_{g}-d\right)/y_{\tau }=\alpha \phi$, hence $2-\tilde{\alpha }=2-\alpha \phi \neq \left(2-\alpha \right)\phi$. This is important to collect correct expressions, and this is shown in Table 2.

This being said, we can now incorporate the logarithmic corrections in Equation (Equation 72) to obtain
(74)$$f_{\mathrm{tri}}^{\mathrm{sing}}\left(\tau ,g,h\right)=b^{-d}F_{\pm }\left(b^{y_{\tau }}{\left(\ln b\right)}^{{\hat{y}}_{\tau }}\tau ,b^{y_{g}}{\left(\ln b\right)}^{{\hat{y}}_{g}}g,b^{y_{h}}{\left(\ln b\right)}^{{\hat{y}}_{h}}h\right),$$
and the comparison with Equation (Equation 71) simply leads to
(75)$$\begin{array}{ccc} & & y_{g}=2,\;y_{h}=\frac{5}{2},\;y_{\tau }=1,\; \end{array}$$
(76)$$\begin{array}{ccc} & & {\hat{y}}_{g}=\frac{1}{3},\;{\hat{y}}_{h}=\frac{1}{6},\;{\hat{y}}_{\tau }=\frac{4}{15},\;\hat{ϙ}=\frac{1}{3}. \end{array}$$

We can now deduce the values of the standard critical exponents and the associated logarithmic corrections exponents. They are listed in Table 3 for three universality classes, which all have non-zero values of the pseudo-critical exponent $\hat{ϙ}$, the Ising model in four dimensions, the tricritical Ising model in three dimensions, and the problem of percolation in six dimensions.

In this table, the first six lines are not controversial. The seventh line presents the correlation function correction exponent $\hat{\eta }$, which follows from our scaling law, in any of the forms given in Equations (Equation 59), (Equation 63), or (Equation 68). These three expressions are mutually consistent, but they differ from Equation (Equation 50) used by Kenna and his coauthors. This latter formula would, respectively, predict for the three universality classes the values 0, $-\frac{1}{3}$, and $\frac{1}{21}$. We have seen that the first value, 0, is falsified in the 4d IM case by Refs. [16,17], but the results of Ref. [18] invalidates our third value $\frac{8}{21}$, the last entry of in Table 3, while Ref. [19], on the contrary, supports this value.

The case of the tricritical Ising model in three dimensions appears crucial, and we have to provide numerical results in support of our result. The numerical computation of the correlation function is known to be a very delicate problem, and we will approach the value of the exponent $\hat{\eta }$ differently, using FSS. Another delicate aspect is also the well-known fact that extracting logarithmic corrections in the vicinity of a critical point can be extremely difficult [25]. Recently, it was found that very accurate results can be obtained numerically in the Blume–Capel model with relatively small system sizes [26] via the analysis of the zeros of the partition function, and in particular the Lee–Yang zeros [27,28]. The Lee–Yang zeros are connected to the susceptibility [29] via
(77)$$\chi \left(g,0,L^{-1}\right)≃L^{-d}\sum_{j=1}^{L^{d}}h_{j}^{-2}\left(g,0,L^{-1}\right),$$
where *j* labels the zeros in the upper-half complex plane, indexed in order of increasing distance from the critical point. The sum is dominated by the lowest zero, the Lee–Yang edge $h^{\mathrm{LY}}$, and at the tricritical point, the FSS of the susceptibility is thus linked to that of $h^{\mathrm{LY}}$:(78)$$\chi \left(0,0,L^{-1}\right)≃L^{-d}h^{\mathrm{LY}}{\left(0,0,L^{-1}\right)}^{-2}≃L^{2-\eta }{\left(\ln L\right)}^{\hat{\eta }}.$$In the presence of logarithmic corrections, the scaling form of the Lee–Yang edge obeys
(79)$$h^{\mathrm{LY}}\left(g,h\right)=b^{-y_{h}}{\left(\ln b\right)}^{-{\hat{y}}_{h}}H_{\pm }\left(b^{y_{\tau }}{\left(\ln b\right)}^{{\hat{y}}_{\tau }}\tau ,b^{y_{g}}{\left(\ln b\right)}^{{\hat{y}}_{g}}g,b^{y_{h}}{\left(\ln b\right)}^{{\hat{y}}_{h}}h\right)$$
compatible with the behavior in terms of the thermal scaling field *g*, as it can be shown using the scaling laws of Table 3, $h^{\mathrm{LY}}\left(g,0\right)∼{\left|g\right|}^{\Delta }{\left(-\ln|g|\right)}^{\hat{\Delta }}$. If one sits exactly at the tricritical point, τ=g=h=0, we can extract the FSS behavior of the zeros by setting b=L,
(80)$$h^{\mathrm{LY}}\left(0,0,L^{-1}\right)≃L^{-y_{h}}{\left(\ln L\right)}^{-{\hat{y}}_{h}}$$
and it follows that we expect
(81)$$h^{\mathrm{LY}}\left(0,0,L^{-1}\right)≃L^{(\eta -2-d)/2}{\left(\ln L\right)}^{-\hat{\eta }/2}$$
which agrees with Equation (Equation 78).

As we said, this form can be checked with good accuracy at the price of relatively light Monte Carlo simulations. The coordinates of the tricritical point of the Blume–Capel model in 3d are found in Ref. [30], $T_{t}≃1.4182$, $\Delta_{t}≃2.8448$, but the value of $\Delta_{t}$ does not seem to be as accurate as that of the temperature and for example Zierenberg et al. Ref [31] report instead $\Delta_{t}≃2.8446$. Let us first analyze this problem ourselves. In Figure 2, we report the FSS of the magnetization at $T_{t}≃1.4182$ for several values of Δ ranging from 2.8440 to 2.8448. The magnetization is expected to follow the FSS behavior
(82)$$m\left(0,0,L^{-1}\right)∼L^{-d+y_{h}}{\left(\ln L\right)}^{{\hat{y}}_{h}}$$
with $d-y_{h}=\frac{1}{2}$ and ${\hat{y}}_{h}=\frac{1}{6}$. The data points are fitted as $m\left(0,0,L^{-1}\right)L^{\frac{1}{2}}=a{\left(\ln L\right)}^{b}$ with a,b free parameters. The closest to the expected result (the black dashed line) is at Δ=2.8442, where we obtain b=0.155±0.007. We will thus consider this value of Δ as our estimate for the coordinate of the tricritical point $\Delta_{t}$.

The analysis of the Lee–Yang edge is presented in Figure 3 with a larger choice of values of Δ and, again, the best fit is at $\Delta_{t}=2.8442$, where the estimate of ${\hat{y}}_{h}$ is now slightly larger at 0.172±0.008.

The reader could still question the sensitivity of the value of ${\hat{y}}_{h}$ with the choice of tricritical temperature $T_{t}$. Indeed, when one looks at the FSS of the tricritical magnetization, for example, the effective exponent of the log term is either positive and close to the expected value or can differ from the expectation and even be negative, depending on the values of the crystal field Δ (see Figure 2). It makes sense to ask whether the role of $T_{t}$ may also have a significant influence. We believe that the results presented in this work are reliable and to support the consistency of the numerical data, we show in Figure 4 that slight variations of *T* change the regime from the pure 3d Ising model at T=1.4197, for which $y_{h}=2.4815(15)$ [32] is expected, to first order at T=1.4170, where an effective FSS $y_{h}=d$ is expected [33]. T=1.4182 safely recovers $y_{h}^{\mathrm{Tri}}=2.5$ to a very good accuracy and confirms the tricritical value of $T_{t}≃1.4182$. Note that the transition line in the phase diagram in the vicinity of the tricritical point is almost at a fixed value of Δ, this is why the three regimes are found at the same crystal-field value of Δ=2.8442.

## 6. Conclusions

The numerical results obtained for the tricritical Ising model universality class in 3d confirm the prediction that ${\hat{y}}_{h}=\frac{1}{6}$, hence the prediction $\hat{\eta }=\frac{1}{3}$, while the scaling law of Kenna and his coworkers would have given $-\frac{1}{3}$ instead.

In Ref. [2], Ralph Kenna concluded his review with a table collecting the sets of critical exponents and hatted critical exponents for various models and predicting those that were still unknown from the use of the newly discovered scaling laws and, in particular Equation (Equation 50), which we scrutinize and propose to replace by Equation (Equation 68) or any of the equivalent forms that we have derived.

In the list, the O(n) model with long-range interactions [34,35] was predicted to have $\hat{\eta }=0$. We propose instead $\hat{\eta }=\frac{1}{2}$, following from ${\hat{y}}_{t}=\left(4-n\right)/\left[2\left(n+8\right)\right]$, ${\hat{y}}_{h}=\frac{1}{4}$.The Lee–Yang edge in 6d [19] was predicted to have $\hat{\eta }=\frac{1}{9}$. We rather have ${\hat{y}}_{t}=-\frac{2}{9}$ and ${\hat{y}}_{h}=\frac{2}{9}$, hence $\hat{\eta }=\frac{4}{9}$.For lattice animals in 8d [19], Kenna predicted $\hat{\eta }=\frac{1}{9}$, and we have ${\hat{y}}_{t}={\hat{y}}_{h}=\frac{2}{9}$ and $\hat{\eta }=\frac{4}{9}$.The case of scale-free networks [36,37,38] is particular in the sense that Ralph Kenna did not make any prediction for $\hat{\eta }$ because some exponents were missing. From those which are known, we can deduce that ${\hat{y}}_{t}=-\frac{1}{2}$ and ${\hat{y}}_{h}=-\frac{1}{4}$, and we deduce thus $\hat{\eta }=-\frac{1}{2}$, which is a new prediction.Eventually, we believe that the *n*-color Ashkin–Teller model in 2d is still under question since the exponents collected by Shalaev and Jug [39] do not satisfy the “standard” scaling laws, e.g., the values reported do not obey $\hat{\alpha }+\hat{\gamma }=2\hat{\beta }$.

To finish this paper, we would like to say that the scaling laws discovered by Ralph Kenna and his coworkers are invaluable because they make it possible to establish (or falsify) the consistency of the results obtained for various models. The case of the *n*-color Ashkin–Teller model in 2d is such an example where it seems that there are still some inconsistencies to solve. Although we happened to contradict one of these scaling laws, we admire the piece of work conducted in Refs. [2,3,4], where we recognize Ralph’s footprint.

## Figures and Tables

**Figure 1 entropy-26-00221-f001:**
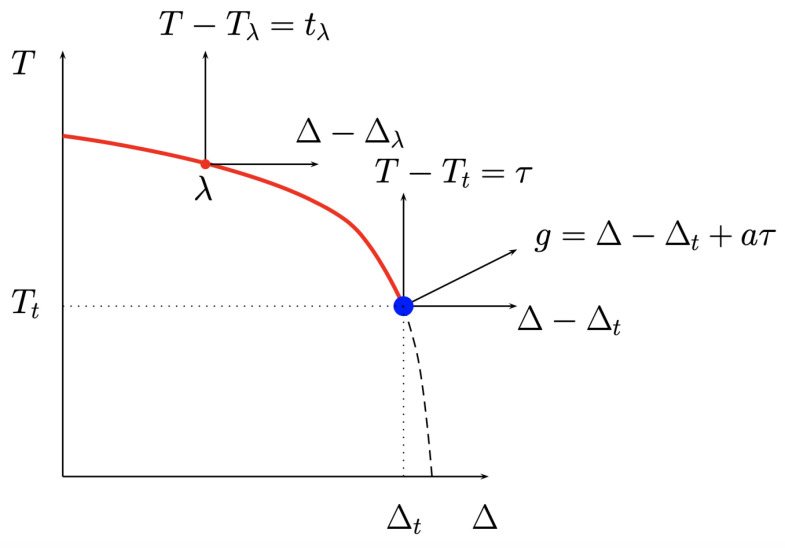
Typical phase diagram of the Blume–Capel model in the (T,Δ) plane. The λ line is a line of second-order phase transition in the Ising model universality class that ends at a tricritical point of coordinates $\left(T_{t},\Delta_{t}\right)$. The dashed line is a first-order transition line. The most singular even scaling field at the tricritical point is *g*, a linear combination of τ and of $\Delta -\Delta_{t}$.

**Figure 2 entropy-26-00221-f002:**
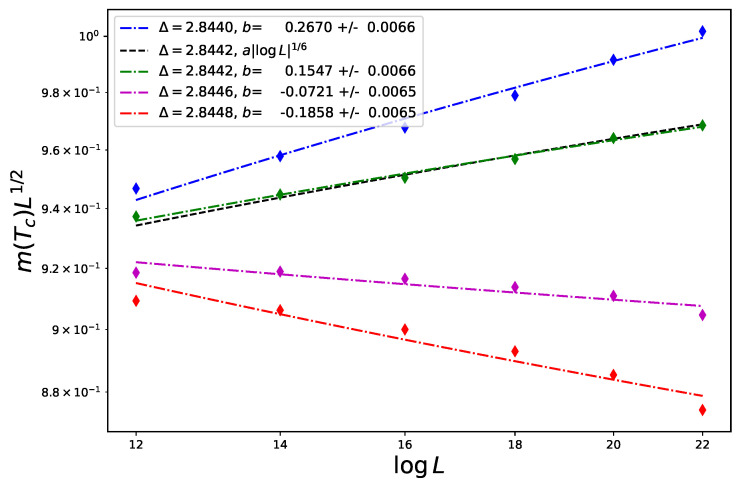
FSS of the magnetization for the tricritical Ising model (Blume–Capel model at its tricritical temperature) in 3d at $T_{t}=1.4182$ and various values of the crystal-field parameter Δ for sizes ranging from L=12 to 22. The best fit is for Δ=2.8442 ($\chi^{2}/\mathrm{dof}=30.40/4=7.6$ at Δ=2.8440, χ2/dof=5.39/4=1.35 at Δ=2.8442, χ2/dof=28.36/4=7.09 at Δ=2.8446 and χ2/dof=76.74/4=19.18 at Δ=2.8448).

**Figure 3 entropy-26-00221-f003:**
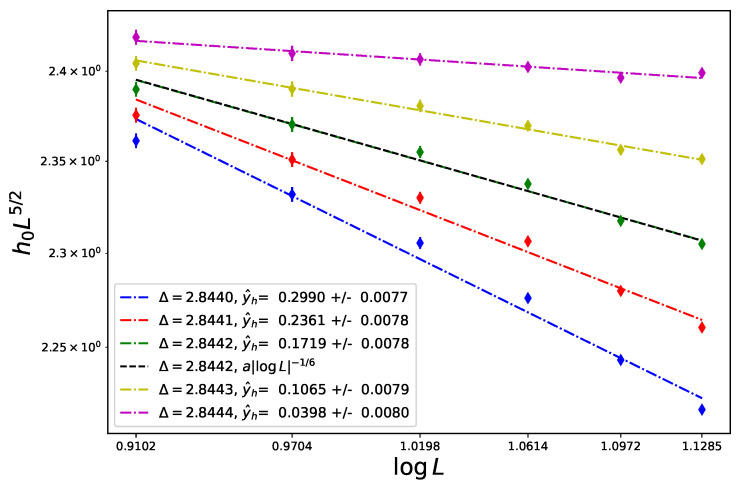
FSS of the Lee–Yang edge for the tricritical Ising model (Blume–Capel model at its tricritical temperature) in 3d at $T_{t}=1.4182$ and various values of Δ for sizes ranging from L=12 to 22.

**Figure 4 entropy-26-00221-f004:**
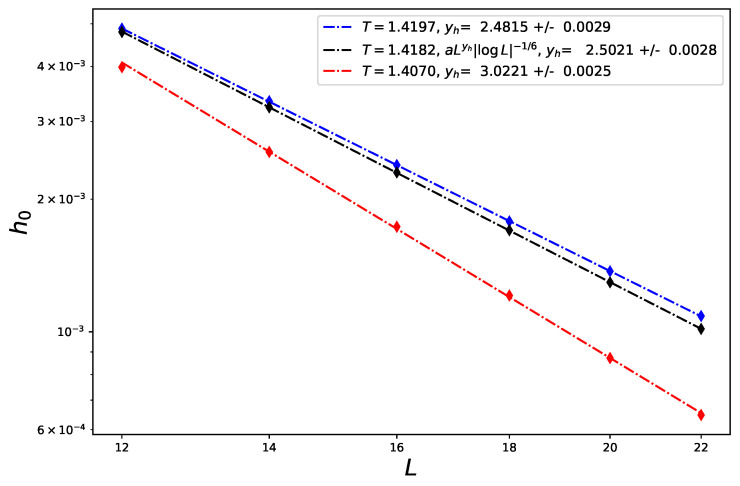
FSS of the Lee–Yang edge in the vicinity of the tricritical point of the Blume–Capel model in 3d, at $T_{t}=1.4170$, 1.4182 and 1.4197 and Δ=2.8442 to show the first-order, the tricritical, and the ordinary second-order regimes from the values of the corresponding RG dimensions $y_{h}$.

**Table 1 entropy-26-00221-t001:** The main scaling laws and hatted scaling laws and the associated universal combinations of amplitudes.

Ratio	Scaling Relation	Hatted Scaling Relation
$\frac{m_{-}^{2}\left(-|t|\right)}{C_{\pm }\left(|t|\right)\chi_{\pm }{\left(|t|)|t\right|}^{2}}$	2β+γ=2−α	$2\hat{\beta }=\hat{\alpha }+\hat{\gamma }$
${\left.\frac{\chi_{\pm }\left(t\right)m_{-}^{\delta -1}\left(-|t|\right)h}{m_{c}^{\delta }\left(h\right)}\right|}_{h=h_{\pm }^{\mathrm{LY}}\left(|t|\right)}$	β(δ−1)=γ	$\hat{\gamma }+\hat{\beta }\left(\delta -1\right)-\delta \hat{\delta }=0$
$\frac{\chi_{\pm }\left(t\right)}{\xi_{\pm }^{2-\eta }\left(t\right){\left(\ln\xi_{\pm }\left(t\right)\right)}^{\hat{\eta }-\gamma \hat{ϙ}/\nu }}$	ν(2−η)=γ	$\hat{\gamma }-\hat{\nu }\left(2-\eta \right)+\gamma \frac{\hat{ϙ}}{\nu }=\hat{\eta }$
$\xi_{\pm }^{d}\left(t\right)f_{\pm }^{\mathrm{sing}}\left(t\right)$	2−α=νd	$\hat{\alpha }=d\hat{ϙ}-d\hat{\nu }$

**Table 2 entropy-26-00221-t002:** Leading and subleading singularities for the most common physical quantities and the definitions of the associated exponents. Here, the crossover exponent is $\phi =y_{g}/y_{\tau }$.

Leading even Field, *g*		Subleading even Field, τ		Leading Odd Field, *h*	
$C\left(0,g,0\right)∼{\left|g\right|}^{-\alpha }$	$\alpha =\frac{2y_{g}-d}{y_{g}}$	$C\left(\tau ,0,0\right)∼{\left|\tau \right|}^{-\tilde{\alpha }}$	$\tilde{\alpha }=\alpha \phi$		
$m\left(0,g,0\right)∼{\left|g\right|}^{\beta }$	$\beta =\frac{d-y_{h}}{y_{g}}$	$m\left(\tau ,0,0\right)∼{\left|\tau \right|}^{\tilde{\beta }}$	$\tilde{\beta }=\beta \phi$	$m\left(0,0,h\right)∼{\left|h\right|}^{1/\delta }$	$\delta =\frac{y_{h}}{d-y_{h}}$
$\chi \left(0,g,0\right)∼{\left|g\right|}^{-\gamma }$	$\gamma =\frac{2y_{h}-d}{y_{g}}$	$\chi \left(\tau ,0,0\right)∼{\left|\tau \right|}^{-\tilde{\gamma }}$	$\tilde{\gamma }=\gamma \phi$		

**Table 3 entropy-26-00221-t003:** Leading and logarithmic correction exponents for the most common physical quantities.

	Leading Exponent				Logarithmic Correction Exponent			
**Quantity**		**IM4D**	**Tri. IM**	**Perco.**		**IM4D**	**Tri. IM**	**Perco.**
C(t,0)	$\alpha =\frac{2y_{t}-d}{y_{t}}$	0	$\frac{1}{2}$	−1	$\hat{\alpha }=\left(2-\alpha \right){\hat{y}}_{t}$	$\frac{1}{3}$	$\frac{1}{2}$	$\frac{2}{7}$
$m_{-}\left(t,0\right)$	$\beta =\frac{d-y_{h}}{y_{t}}$	$\frac{1}{2}$	$\frac{1}{4}$	1	$\hat{\beta }=\beta {\hat{y}}_{t}+{\hat{y}}_{h}$	$\frac{1}{3}$	$\frac{1}{4}$	$\frac{2}{7}$
χ(t,0)	$\gamma =\frac{2y_{h}-d}{y_{t}}$	1	1	1	$\hat{\gamma }=2{\hat{y}}_{h}-\gamma {\hat{y}}_{t}$	$\frac{1}{3}$	0	$\frac{2}{7}$
$m_{c}\left(0,h\right)$	$\frac{1}{\delta }=\frac{d-y_{h}}{y_{h}}$	$\frac{1}{3}$	$\frac{1}{5}$	$\frac{1}{2}$	$\hat{\delta }=\frac{1}{\delta }{\hat{y}}_{h}+{\hat{y}}_{h}$	$\frac{1}{3}$	$\frac{1}{5}$	$\frac{2}{7}$
$h_{\mathrm{LY}}\left(t,0\right)$	$\Delta =\frac{y_{h}}{y_{t}}$	$\frac{3}{2}$	$\frac{5}{4}$	2	$\hat{\Delta }=\Delta {\hat{y}}_{t}-{\hat{y}}_{h}$	0	$\frac{1}{4}$	0
ξ(t,0)	$\nu =\frac{1}{y_{t}}$	$\frac{1}{2}$	$\frac{1}{2}$	$\frac{1}{2}$	$\hat{\nu }=\hat{ϙ}-\nu {\hat{y}}_{t}$	$\frac{1}{6}$	$\frac{1}{6}$	$\frac{5}{42}$
G(0,0,|r|)	$\eta =d-2y_{h}+2$	0	0	0	$\hat{\eta }=2{\hat{y}}_{h}$	$\frac{1}{2}$	$\frac{1}{3}$	$\frac{8}{21}$

## Data Availability

No new data were created or analyzed in this study. Data sharing is not applicable to this article.

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
