# Peer review of "Ralph Kenna’s Scaling Relations in Critical Phenomena"

_entropy, 2024, doi:10.3390/e26030221_

Round 1

Reviewer 1 Report

Comments and Suggestions for Authors

The paper provides a correction to one of the scaling laws for lograithmic exponents earlier suggested by Kenna et.al. in references [2,3,4].

In place of the posited eq.(50) in the earlier work an alternative scaling relation is derived (eq.(68) or equivalent versions). The key difference is the additional term described in lines 162-163. As the authors point out the original scaling relations are consistent with analytical predictions for percolation in 6d in [18] but, importantly, not with the FSS results in [19] - which rather support the new predicted values. The modified scaling relation also obtains \hat{\eta}=1/2 for the 4d Ising model, which appears to be the correct value.

Further evidence is support of the modified scaling law is presented for the tricritical Ising model universality class, where a positive \hat{\eta} value of 1/3, in agreement with the "new" scaling relation,  is measured rather than the negative value of -1/3 predicted by the "old" scaling relation. Although it is notoriosly difficult to extract logarithmic exponents numerically the methods employed by the authors look convincing.

The list of other models in the conclusions where the two variant scaling relations give different results for exponents is useful and suggests some potential directions for future study. 

Only one minor comment on formatting - the indentations in lines 323-334 where various exponent predictions are listed look a little odd. Were they intentional? There are also one or two typos, e.g. firt instead of first in the caption of Figure 4.

Reviewer 2 Report

Comments and Suggestions for Authors

The report  recalls  the research way of research, who notaby influnced the Critical Phenomena topic. In my opinion it should be publised, and it can be published in the current form. For future reader such resume cn be important.

Reviewer 3 Report

Comments and Suggestions for Authors

It is always a pleasure to read well prepared papers. I haven't found any issues to be corrected. Maybe some parts could be less/more general. However, describing critical phenomena theory the Authors had to decide which part should be emphasized. I suggest accept this paper in its present form.